# Identifying Disease Signatures in the Spinocerebellar Ataxia Type 1 Mouse Cortex

**DOI:** 10.3390/cells11172632

**Published:** 2022-08-24

**Authors:** Kimberly Luttik, Victor Olmos, Ashley Owens, Aryaan Khan, Joy Yun, Terri Driessen, Janghoo Lim

**Affiliations:** 1Interdepartmental Neuroscience Program, Yale School of Medicine, New Haven, CT 06510, USA; 2Department of Neuroscience, Yale School of Medicine, New Haven, CT 06510, USA; 3Department of Genetics, Yale School of Medicine, New Haven, CT 06510, USA; 4Yale College, New Haven, CT 06510, USA; 5Program in Cellular Neuroscience, Neurodegeneration and Repair, Yale School of Medicine, New Haven, CT 06510, USA; 6Yale Stem Cell Center, Yale School of Medicine, New Haven, CT 06510, USA

**Keywords:** spinocerebellar ataxia type 1, SCA1, neurodegeneration, regional vulnerability, cortex

## Abstract

The neurodegenerative disease spinocerebellar ataxia type 1 (SCA1) is known to lead to the progressive degeneration of specific neuronal populations, including cerebellar Purkinje cells (PCs), brainstem cranial nerve nuclei and inferior olive nuclei, and spinocerebellar tracts. The disease-causing protein ataxin-1 is fairly ubiquitously expressed throughout the brain and spinal cord, but most studies have primarily focused on the role of ataxin-1 in the cerebellum and brainstem. Therefore, the functions of ataxin-1 and the effects of SCA1 mutations in other brain regions including the cortex are not well-known. Here, we characterized pathology in the motor cortex of a SCA1 mouse model and performed RNA sequencing in this brain region to investigate the impact of mutant ataxin-1 towards transcriptomic alterations. We identified progressive cortical pathology and significant transcriptomic changes in the motor cortex of a SCA1 mouse model. We also identified progressive, region-specific, colocalization of p62 protein with mutant ataxin-1 aggregates in broad brain regions, but not the cerebellum or brainstem. A cross-regional comparison of the SCA1 cortical and cerebellar transcriptomic changes identified both common and unique gene expression changes between the two regions, including shared synaptic dysfunction and region-specific kinase regulation. These findings suggest that the cortex is progressively impacted via both shared and region-specific mechanisms in SCA1.

## 1. Introduction

Spinocerebellar ataxia type 1 (SCA1) is an inherited neurodegenerative disorder characterized by progressive gait impairment, limb incoordination, respiratory dysfunction, cognitive impairment, and memory decline [1,2,3,4,5,6,7]. SCA1 is a monogenic disorder caused by a CAG repeat expansion in the gene *ATAXIN-1* (*ATXN1*), resulting in a polyglutamine (polyQ) expansion in the ataxin-1 protein [8]. Interestingly, the disease-causing gene *ATXN1* is fairly ubiquitously expressed throughout the central nervous system [9], but is known to lead to pronounced neuron loss and pathology in specific brain regions, including the cerebellum and brainstem [10,11,12]. However, pathology and molecular disease signatures in other brain regions have not yet been fully understood. 

The pronounced cerebellar Purkinje cell (PC) loss and ataxic gait impairment have led to impactful SCA1 research within the cerebellum and brainstem. Despite large focus in these regions, analysis of SCA1 patient brains by magnetic resonance imaging (MRI) and post-mortem imaging has identified broad degeneration and pathology in both cerebellar and extra-cerebellar regions, including the thalamus, striatum, and frontal, temporal, and parietal lobes of the cortex [11,12,13,14]. In addition, similar to other neurodegenerative diseases [15,16,17,18,19,20,21,22], aggregates of the disease-causing protein ataxin-1 are broadly observed across brain regions in SCA1 mouse models, including the cortex, striatum, and to a lesser degree in the cerebellum [10]. The functional impacts of mutant ataxin-1 expression and aggregation in extra-cerebellar regions are not fully understood [23]; specifically, whether this expression results in similar or different degrees of disease progression across brain regions, as well as how this might contribute to symptoms experienced by patients, including memory decline, executive dysfunction, and cognitive impairment [4,5,6,7]. Previous studies performing transcriptomic analyses of SCA1 mouse tissues, including cerebellum and brainstem [24,25,26,27,28], have provided important insight into region-specific and shared transcriptomic changes, which may aid in identifying brain-region specific disease pathways and eventually developing more effective therapeutics. 

In this study, we aimed to understand the functional impact of mutant ataxin-1 expression in extra-cerebellar regions and identify potential molecular mechanisms underlying observed clinical phenotypes, including cognitive impairment. To do this, we utilized the SCA1 knock-in (KI) mouse model, which expresses *Atxn1* [154Q] under its endogenous regulators, and replicates many key aspects of SCA1 disease progression and pathology, including progressive behavioral and cognitive impairments, as well as mutant ataxin-1 aggregation, gliosis, and PC pathology [10]. First, we characterized cortex-specific pathological and molecular changes in the SCA1 KI mouse model. We observed progressive cortical thinning in the SCA1 KI motor cortex, and progressive, region-specific, increases in p62-positive aggregates that colocalize with ataxin-1 nuclear inclusions in a polyQ-dependent manner. Next, we analyzed disease-specific molecular signatures in the cortex and identified a large degree of transcriptomic changes in the SCA1 KI mouse cortex. Lastly, we performed regional transcriptomic comparisons between the SCA1 KI mouse cortex and cerebellum, and found both shared (e.g., synaptic dysfunction) and region-specific differences (e.g., kinase regulation) in pathways altered during early stages of SCA1 disease progression. Collectively, these data indicate that the cortex is affected in the SCA1 KI mouse model, and identifies potential shared and unique mechanisms in which cortex and cerebellum are impacted in SCA1.

## 2. Materials and Methods

### 2.1. Animal Husbandry

All animal care procedures were appropriately approved by the Yale University Institutional Animal Care and Use Committee (IACUC; Approval Code: 2021-11342, Approval Date: 21 December 2021). Mice were kept in a 12 h light/dark cycle with standard chow and ad libitum access to water. The SCA1 KI (*Atxn1*^154Q/2Q^) [10] mouse strain was utilized for all experiments, which expresses mutant ataxin-1 with 154 glutamine repeats under its endogenous regulators, maintained on a pure C57BL/6J background. SCA1 KI and wild-type (WT) littermate controls aged to 12 and 30 weeks, with a combination of males and females, were used for all experiments.

### 2.2. Cell Culture

Neuro-2a (N2A; ATCC, CCL-131) cells were plated at 5 × 10^4^ cells per well in a 24 well plate with a cover slip 24 h prior to transfection, maintained in a 37 °C, 5.5% CO_2_ incubator and cultured in DMEM (Gibco) with 10% FBS, 100 U/mL penicillin, and 100 U/mL streptomycin. When 70–80% confluent, media was replaced with antibiotic free DMEM with 10% FBS, and cells were transiently transfected with 0.5 ug DNA, 1 uL P3000 enhancer reagent (Invitrogen, L3000015, Waltham, MA, USA), and 1 uL Lipofectamine 3000 (Invitrogen, L3000015) in 50 uL Opti-MEM (Gibco). After 48 h, cells were washed twice with PBS and subsequently fixed with 4% PFA for 10 min. Cells were washed two more times with PBS, prior to blocking for 1 h in 10% normal goat serum and 0.3% Triton-X in PBS. Cells were then incubated at 4 °C in primary antibody in blocking buffer overnight (chicken anti-DDDDK (Abcam, ab1170, 1:400, Cambridge, UK), mouse anti-p62 (SQSTM1) (Abnova, H00008878-M01, 1:700, Taipei City, Taiwan)). The following morning, cells were washed three times in PBS prior to incubation in secondary antibody in blocking buffer for two hours at room temperature (goat anti rabbit-488 and goat anti mouse-555 (Invitrogen, 1:500)). Following incubation, cells were washed three times in PBS and mounted on slides using Vectashield mounting media and DAPI (Vector Laboratories, H-1500, Newark, NJ, USA). Fluorescent images were acquired on a Zeiss LSM880 confocal microscope at 20× magnification, with the same settings used across similar experiments. Images were acquired and quantified for 50 cells per condition.

### 2.3. Protein Extraction and Western Blot Analysis

Protein extraction and Western blot analysis of whole mouse cerebellar and cortical tissue was performed as previously described [29]. The following primary antibodies were used: mouse anti-Gapdh (Sigma G8795, 1:10,000, St. Louis, MO, USA), mouse anti-Vinculin (Millipore v9264 1:10,000, Burlington, MA, USA), rabbit anti-Gfap (Sigma G4546, 1:500), rabbit anti-Iba1 (Wako, 019-19741, 1:400), rabbit anti-Prkcb (Thermo Fisher Scientific 12919-1-AP, 1:1000, Waltham, MA, USA), rabbit anti-Camkk2 (Thermo Fisher Scientific 11549-1-AP, 1:1000).

### 2.4. RNA Extraction and RT-qPCR

RNA extraction and reverse transcription-quantitative polymerase chain reaction (RT-qPCR) was performed as previously described [29]. The following TaqMan (Applied Biosystems, Waltham, MA, USA) probes were used: *Gapdh* (4352661, Mm99999915_g1), *Actb* (4352933 E, Mm00607939_s1), *Hprt* (4331182, Mm03024075_m1), *Camkk2* (4331182, Mm00520236_m1), and *Prkcb* (4331182, Mm00435749_m1). Target gene expression was normalized to housekeeping genes (*Actb*, *Gapdh*, and *Hprt*) using BioRad CFX manager software and plotted relative to mean expression of controls. 

### 2.5. RNA Extraction and Bulk RNA Sequencing

Whole mouse cerebella and cortex were macro-dissected, flash frozen, and stored at −80 °C until processing. RNA was extracted using the Qiagen RNeasy Mini Kit, and DNA was removed with DNase I as described in the manufacturer’s protocol. Total RNA was sent to the Yale Center for Genome Analysis (YCGA) for processing. RNA integrity was measured with capillary electrophoresis (Agilent BioAnalyzer 2100, Agilent Technologies, Santa Clara, CA, USA) and RIN values were checked to ensure RNA integrity before proceeding with RNA-seq. Libraries were generated using oligo-dT purification of poly-adenylated RNA, which was then reverse transcribed into cDNA that was fragmented, blunt ended, and ligated to adaptors. Base pair size was determined with capillary electrophoresis, and the library was quantified before pooling and sequencing on a HiSeq 2500 using a 75 bp paired-end read strategy. Sequencing data can be accessed under GEO accession # (GSE211678).

### 2.6. RNA Sequencing Data Analysis

STAR (Spliced Transcripts Alignment to a Reference) [30] was utilized to align FASTQ reads to mouse reference genome (Mm10). FeatureCounts [31] was used to quantify read counts per gene in Python 3.10. The obtained read counts were used for subsequent quantification and differential expression analysis, completed using the DESeq2 package [32] in the R environment. Gene expression with an FDR-adjusted *p*-value < 0.05 was considered significant, and genes with a Log2FoldChange greater than |0.25| were used in the final analyses. The Pheatmap (RRID:SCR_016418) and BioVenn [33,34] packages were used to generate all heatmaps and Venn diagrams, respectively, in the R environment (Version 4.1.3, the r-project). Normalized read counts were used as heatmap input. Intervene [35] was used to generate Upset plots. Gene Ontology analysis was conducted using g: Profiler [36,37]. Plots were generated using GraphPad Prism 9 (Version 9.3.1, GraphPad Software, San Diego, CA, USA) and Python 3.10 (Python Software Foundation, Wilmington, DE, USA).

### 2.7. Immunohistochemistry

Immunohistochemistry was performed as previously described [29]. The following primary antibodies were used: mouse anti-NeuN (Sigma Aldrich, MAB327, 1:800), rabbit anti-Iba1 (Wako, 019-19741, 1:500), rabbit anti-ataxin-1 [11NQ] (1:500), mouse anti-p62 (SQSTM1) (Abnova, H00008878-M01, 1:700). Primary motor cortex (MOp) was imaged based on Allen Institute for Brain Science mouse reference atlas [38]. Between 3–6 fluorescent images were acquired per brain region per mouse on a Zeiss LSM880 confocal microscope at 10× or 20× magnification, with the same settings used across similar experiments. 

### 2.8. Fluorescent Image Quantification

ImageJ (National Institutes of Health) was used for all image processing and quantification. For all images across experiments, the maximum intensity projection was utilized, converted to 8-bit, and thresholded using identical parameters across all images of each experiment. ROI manager was used to measure equal areas across images. To quantify cortical thickness, the length of the cortical layer was measured at three locations in the image approximately 100 μm apart, from layer 1 to layer 6 based on NeuN stain, and averaged per image. To quantify neuron number and size, clumped nuclei were split using the watershed function, and only NeuN-positive nuclei greater than 30 μm^2^ were counted. To quantify microglia number and intensity, only Iba1-positive cells greater than 15μm^2^ were counted, and intensity was recorded as mean gray value. To quantify p62-positive puncta per mm^2^, only puncta greater than 3μm^2^ were counted. To quantify colocalization of p62-positive puncta with ataxin-1 nuclear inclusions (ataxin-1 [11NQ] antibody-specific), the ImageJ plugin JaCoP was used [39]. Images were cropped to contain a single cell of interest and thresholded to remove background intensity. Percent colocalization was then calculated using the Manders’ overlap coefficient. For all quantifications of mouse tissue, 3–6 images were quantified per mouse, and for quantifications of in vitro cells, 50 cells per condition were quantified. GraphPad Prism was used to plot and analyze all fluorescent image data. 

### 2.9. Statistical Analysis

All statistical analyses were completed using GraphPad Prism software. All data are shown as mean ± SEM. When comparing two experimental groups, a two-tailed unpaired student’s *t* tests was performed. When comparing more than two experimental groups, a one-way analysis of variance (ANOVA) with Tukey’s post hoc analysis was performed. Statistical significance was determined using a significance cutoff of *p* < 0.05. 

## 3. Results

### 3.1. Progressive Cortical Thinning in the SCA1 KI Mouse Model

In order to identify region-specific disease signatures between the SCA1 motor cortex and cerebellum, we first aimed to broadly investigate the degree in which the cortex might be affected or altered in mouse models of SCA1. It is known that ataxin-1 is expressed in the cortex [9], and that polyQ-expanded mutant ataxin-1 aggregates in nuclear inclusions across brain regions in SCA1 mouse models, including the cortex [10]. However, the functional impacts of this expression and aberrant aggregation in extra-cerebellar regions are not fully understood. We analyzed SCA1 KI mouse brain tissue at 12 and 30 weeks, as these are known to represent mid- and late stages of SCA1 disease progression in the mouse cerebellum, and are time points at which motor and cognitive impairment is observed in SCA1 KI mice [10,40,41]. At 12 weeks, there is significant astro- and micro-gliosis observed in the cerebellum [40], as well as motor and cognitive behavioral deficits, including impaired spatial learning by Morris Water Maze, and impaired learning and memory by conditioned fear response [10,41]. At 30 weeks, these phenotypes are more severe, and cerebellar PC loss is observed as well [10,41].

As cortical thinning has been observed in studies of SCA1 patients [12,42], we first looked at whether this was replicated in the SCA1 KI mouse model. We confirmed progressive cortical thinning in the SCA1 KI mouse motor cortex, with trending changes at 12 weeks (early stage of disease progression), and a significant decrease at 30 weeks (late stage of disease progression), compared to WT littermate controls (Figure 1A,B). To identify whether this may be due to neuronal loss or atrophy, we characterized the density and size of NeuN-positive neuronal nuclei in the mouse cortex. We observed no significant neuronal loss at 12 or 30 weeks (Figure 1A,C), but did observe a progressive decrease in size of neuronal nuclei, with a significant decrease at 30 weeks in both layers 2/3 (L2/3) and layer 5 (L5) of the SCA1 KI motor cortex (Figure 1A,D). This could suggest that the observed cortical thinning may be due to atrophy, not necessarily death, of cortical neurons. However, this does not necessarily exclude the possibility of neuronal subpopulations being differentially affected in the cortex. 

Next, since gliosis is reported in the SCA1 KI cerebellum at these time points [40], we investigated gliosis in the motor cortex. We observed no significant changes in microgliosis in Iba1-positive populations at 12 or 30 weeks (Figure 1E–K), although we did observe a trending increase in Iba1-positive microglia number at 12 weeks (Figure 1E,G). We also did not observe any significant changes in protein levels of astrocyte marker Gfap at 12 weeks (Figure 1J,L). Interestingly, we previously analyzed gliosis in the SCA1 KI mouse cerebellum and inferior olive, and observed significant astro- and micro-gliosis at 12 weeks in the inferior olive [27]. This suggests that there are differential patterns of gliosis across brain regions in SCA1. As we only analyzed a subset of astrocyte and microglia markers in this study, this does not necessarily exclude the possibility of glial subpopulations being differentially impacted in the SCA1 cortex. Taken together, these data suggest that the cortical thinning observed in SCA1 patients is replicated in an age-dependent manner in the SCA1 KI mouse model, and that this thinning may in part be due to neuronal shrinkage or atrophy. Interestingly, the gliosis observed in the SCA1 KI cerebellum and inferior olive at 12 and 30 weeks [27,40] was not significantly changed in the SCA1 KI cortex at these same time points. 

### 3.2. Progressive, Region-Specific, PolyQ-Dependent Colocalization of p62 with Mutant Ataxin-1 Nuclear Inclusions

Next, we were interested in investigating the ataxin-1 nuclear inclusions in the cortex. Aggregations of disease-causing proteins are observed across diverse neurodegenerative disorders [15,16,17,18,19,20,21,22]; however, the impact of these aggregates towards disease progression is still not fully understood. In SCA1, ataxin-1-positive aggregates are observed across diverse brain regions, although less so in the cerebellum [10]. Here, we imaged ataxin-1 aggregates in the motor cortex and observed that these aggregates colocalized with the autophagy/proteaseome marker p62 (Figure 2A–C). Interestingly, p62 aggregates were progressively increased in the SCA1 cortex, with significant increase compared to WT littermate controls at both 12 and 30 weeks (Figure 2A–C). We also found that colocalization of p62-positive aggregates and ataxin-1 nuclear inclusions was observed in other brain regions of SCA1 mice, including the striatum and hippocampus, with minimal to no colocalization in the SCA1 brainstem and cerebellum at 30 weeks (Appendix A). Given the role of p62 in autophagy and proteasomal degradation of ubiquitinated proteins [43], as well as previous literature suggesting that p62 may aid in degrading aggregated polyQ-expanded ataxin-1 in vitro [44], we were interested if this colocalization occurred in a polyQ-dependent manner. To investigate this, we transfected N2A cells with 2Q, 30Q, and 82Q human *ATXN1* (Flag-*hATXN1*) plasmids (Figure 2D). After 48 h, 44.3% of N2A cells transfected with *hATXN1* [82Q] formed mutant ataxin-1 nuclear inclusions colocalized with endogenous p62, in contrast with 9.9% and 11.3% of N2A cells transfected with *hATXN1* [2Q] and −[30Q] having p62-positive ataxin-1 nuclear inclusions, respectively (n = 50 cells per condition) (Figure 2D,E). This significant increase in *hATXN1* [82Q] colocalization with p62 in nuclear inclusions, along with the observed increase in p62 aggregates in SCA1 cortex in vivo, suggests that this colocalization occurs in a polyQ-dependent and region-specific manner. 

### 3.3. Transcriptomic Alterations in the SCA1 KI Mouse Cortex 

Given these observed morphological changes in the SCA1 KI mouse cortex, we were interested in investigating potential molecular mechanisms in which mutant ataxin-1 impacts the cortical transcriptome, as the role of ataxin-1 as a transcriptional regulator, and the impact of this towards the transcriptional signature of the cerebellum and inferior olive, has been well established [8,24,25,26,27,45]. We performed bulk RNA sequencing of the SCA1 KI mouse cortex at 12 weeks, and observed a high number of up- and downregulated genes in the SCA1 cortex compared to WT littermate controls (1853 downregulated differentially expressed genes (DEGs), 819 upregulated DEGs) (Figure 3A,B, Appendix A). Genes with an FDR-adjusted *p*-value < 0.05 and a log_2_foldchange (LFC) greater than |0.25| were considered significant and used for further analyses. The top Gene Ontology (GO) terms for the downregulated DEGs were largely related to phosphorylation and kinase regulation, calcium channel activity, synaptic organization and function, and dendrite development, etc. (Figure 3C,D, Appendix A). In contrast, the top GO terms for the upregulated DEGs were mainly related to transcriptional regulation and chromatin binding (Appendix A). Together, these data support that mutant ataxin-1 expression is significantly impacting the cortical transcriptome in the SCA1 KI mice.

### 3.4. Cross-Tissue Comparison of the Cortex and Cerebellum Identifies Shared and Unique Transcriptomic Response to SCA1 Disease

Given the significant degree of transcriptomic alterations in the SCA1 KI mouse cortex, we were interested in seeing how this compares to that of the SCA1 KI mouse cerebellum. To do this, we reanalyzed our lab’s previously published cerebellar transcriptomic dataset [27] and compared this to the cortical transcriptomic dataset (Figure 4). Surprisingly, we observed a larger degree of DEGs in the cortex at 12 weeks compared to the cerebellum, with 2672 DEGs in the cortex (819 upregulated, 1853 downregulated), and 1154 DEGs in the cerebellum (232 upregulated, 922 downregulated) at this time point (Figure 4B, Appendix A). Of the upregulated DEGs, there was a 20.26% or 5.74% (n = 47) overlap of the cerebellar DEGs (n = 47/232) with cortical DEGs (47/819), respectively (Figure 4C,D). For the downregulated DEGs, this overlap was 30.37% or 15.11% (n = 280) of cerebellar DEGs (n = 280/922) with cortical DEGs (n = 280/1853), respectively, suggesting these genes are commonly altered across these brain regions, with the remainder being unique to each region (Figure 4C–E and Appendix A). Interestingly, there were a subset of DEGs that behaved in an opposing manner between these two brain regions, showing upregulation in the cortex but downregulation in the cerebellum (n = 54), and vice versa (n = 45) (Figure 4C). Together, these data suggest that mutant ataxin-1 impacts gene expression in both the cerebellum and cortex, with a larger degree of transcriptomic changes observed in the SCA1 KI mouse cortex compared to the cerebellum at 12 weeks. 

Next, we were interested in understanding what biological and molecular pathways were commonly or uniquely altered across the cerebellum and cortex, as this may suggest common or distinct disease signatures across brain regions. Comparison of the SCA1 KI mouse cerebellar and cortical transcriptomic changes at 12 weeks identified both shared and unique GO hits (Figure 4F and Appendix A). The top shared GO hits that were downregulated in both the cerebellum and cortex related to synaptic transmission, cell to cell signaling, and nervous system development, etc. (Figure 4F, Appendix A). This suggests that synaptic dysfunction is a shared common disease signature in SCA1 KI mice across these regions at 12 weeks. In contrast, the top shared GO hits that were upregulated in both the cerebellum and cortex included system development, neuron differentiation, and regulation of macromolecule metabolic process, etc. (Appendix A).

We also identified region-specific differences in transcriptional alterations. We found that the top GO hits for downregulated DEGs that were unique to the cerebellum related to G-protein coupled receptor signaling, dopamine response, and serotonin receptor signaling, etc. (Figure 4F, Appendix A). In contrast, kinase regulation and phosphorylation were consistently downregulated in the SCA1 KI mouse cortex, but unchanged in the SCA1 cerebellum (Figure 4F, Appendix A). Among the upregulated DEGs, cell morphogenesis and transcription regulation were specific to the SCA1 KI mouse cerebellum and the cortex, respectively (Appendix A). Given previous literature investigating kinases in SCA1 [46,47,48], including region-specific targeting of kinases as a potential effective therapeutic approach [48], we further investigated this change in gene expression. We confirmed a significantly altered expression of serine-threonine kinases in the SCA1 KI mouse cortex that was not observed in the cerebellum (Figure 5A). We validated that this downregulation was observed only in the SCA1 KI mouse cortex and not the cerebellum at 12 weeks for a subset of kinases, specifically *Camkk2* and *Prkcb*, as these were among the top 10 differentially expressed kinases in the cortex (Figure 5B–D). Taken together, these data suggest that there are region-specific and shared cross-regional molecular changes occurring in the SCA1 KI mouse brain (Figure 4, Figure 5 and Figure 6, Appendix A). 

## 4. Discussion

In an effort to further understand the functional impact of mutant ataxin-1 expression in broad brain regions in SCA1, we characterized cortex-specific changes in the SCA1 KI mouse model as this region has been largely understudied, and performed a cross-regional comparison of the transcriptomic changes between brain regions of the cortex and cerebellum. We identified region-specific pathological and molecular changes in SCA1, including progressive cortical thinning, protein aggregation and colocalization of p62 with ataxin-1 nuclear inclusions, as well as common and cortex-specific transcriptomic alterations, including regulation of kinase activity.

Importantly, we observed progressive cortical shrinkage in the SCA1 KI mouse motor cortex, which is consistent with the cortical thinning observed in human SCA1 patients [12,42]. Our data suggest that this shrinkage may be due to neuronal atrophy, rather than significant neuronal loss even at 30 weeks in SCA1 KI mice. Further research investigating the progression of this phenotype at later time points would be insightful, as well as investigation into greater detail of how unique cortical neuron subtypes might be differentially affected during SCA1 disease initiation and progression, as certain populations may have differential responses to mutant ataxin-1 expression compared to others. Further, we did not observe significant astro- or micro-gliosis in the SCA1 motor cortex at 12 or 30 weeks, although we saw a trending increase in microglia density at 12 weeks. Interestingly, gliosis is observed in the SCA1 KI mouse cerebellum and inferior olive at these time points [27,40]. Region-specific differences in gliosis [27], and the functional impact of gliosis towards neuronal health and pathology is an interesting area of future study. It is worth mentioning that our data cannot exclude the possibility of glial subtypes being differentially affected in the cortex, as our findings are based on pan-astrocyte (Gfap) and microglia (Iba1) markers. Additionally, in this study we investigated the SCA1 mouse motor cortex, primarily layers 2/3 and 5. Thus, it is possible that other cortical areas and layers may have diverse or different responses to mutant ataxin-1 expression. 

We observed differential region-specific aggregation and colocalization of p62 with ataxin-1 nuclear inclusions. Colocalization of p62 aggregates with ataxin-1 nuclear inclusions was observed in the SCA1 KI mouse cortex, hippocampus, and striatum, but not in the cerebellum or brainstem. Interestingly, this colocalization occurred in a polyQ-dependent manner in N2A cells in vitro. The functional impact of this colocalization is of interest, especially given that the regulation and role of p62 has been quite well studied in other neurodegenerative disease, including Alzheimer’s disease (AD) [49], Parkinson’s disease (PD) [50,51], Huntington’s disease (HD) [52], amyotrophic lateral sclerosis (ALS), spinocerebellar ataxias (SCAs) [44,53], as well as various cancers [54,55,56]. P62, encoded by *SQSTM1*, is a scaffold protein known to promote degradation of ubiquitinated proteins, and therefore thought to have protective functions by promoting clearance of protein aggregates [54,57]. Therapeutic compounds targeting autophagy, including rapamycin and metformin [54,58,59,60,61,62,63,64,65], have shown some promise in neurodegenerative diseases. Rapamycin that inhibits mTOR and enhance autophagy has been reported to be beneficial in models of AD [59,60,61], HD [62,63,64], but detrimental in ALS models [65]. In SCA1 specifically, an in vitro study found that p62 played a positive role in regulating clearance of mutant ataxin-1 protein [44]. In addition to autophagy, p62 is also known to play roles in nucleo-cytoplasmic shuttling, regulation of inflammation, and oxidative stress [54,55,66,67]. Given this wide range of functions of p62, and its progressive, region-specific distribution in the SCA1 KI mouse model, further investigation into the role of p62 in SCA1 progression would be of interest. 

Our bulk RNA sequencing of the whole SCA1 KI mouse cortex was performed at 12 weeks, as this is considered to be an early to-intermediate stage of disease progression in SCA1 KI mice, with motor and cognitive impairments observed, as well as gliosis and nuclear inclusions, but no major cellular loss in the cerebellum [10,27,40,41]. We selected an early to-intermediate stage rather than an end-stage of disease progression in order to avoid any potential distorted transcriptional alterations that may be due to substantial cell loss. Further, this 12-week time point aligns with previously generated bulk RNA sequencing data of the SCA1 cerebellum [27], and allowed for a cross-regional comparison between these two brain regions. Our bulk RNA sequencing analyses of the whole SCA1 KI mouse cortex identified a large number of transcriptomic changes occurring in the cortex at 12 weeks, including hits relating to synapse function, transcription regulation, and kinase activity, etc. The functional impact of these changes, whether they are pathogenic, protective, or secondary to disease progression still requires further investigation. Interestingly, in our cross-regional comparison between the cortex and cerebellum, we observed a larger degree of transcriptomic changes in the cortex compared to the cerebellum, with a subset of genes overlapping across brain regions. This suggests that there are significant changes occurring in the SCA1 KI cortex, and how these changes are involved in disease progression is an important area of further investigation. To understand how these two brain regions might be differentially or similarly impacted in SCA1, we compared the top GO hits across cortex to cerebellum, and observed shared GO hits relating to synaptic transmission and cell to cell signaling, suggesting that these may be commonly affected across brain regions in SCA1. In contrast, dysregulation, in particular downregulation, of serine-threonine kinases was observed in the SCA1 KI cortex, with no significant changes in the cerebellum. The functional impact of this remains unknown; however, it is of particular interest given previous research in SCA1, showing inhibition or reduction in several kinases to be sufficient to rescue pathology in the cerebellum and brainstem [46,47,48]. Once again, it remains unclear at this moment what the functional impact of kinase reduction in the cortex is; for example, whether this is pathogenic, compensatory, or neutral to SCA1 pathology. 

Lastly, our present study describes pathological and molecular changes in the SCA1 KI mouse cortex and clearly suggests that the cortex is indeed affected in this SCA1 mouse model. Importantly, SCA1 patients also experience a wide range of symptoms related to cortex. These include cognitive impairment and memory decline, specifically executive dysfunction, impaired verbal memory, and decreased visuospatial performance, which tends to correlate with ataxia symptoms [4,5,6,7]. The functional impact of these cortex-specific and shared molecular or pathological changes, and how these may contribute towards clinical symptoms in patients, remain an important area of future studies, as this may aid in developing more effective therapeutic targets to improve patient outcomes. 

## Figures and Tables

**Figure 1 cells-11-02632-f001:**
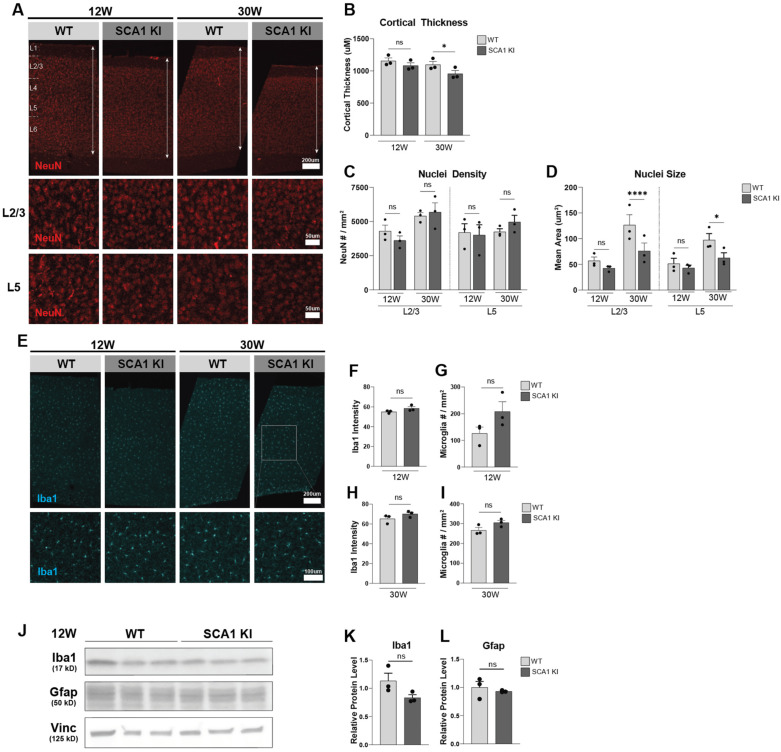
Progressive cortical thinning in the SCA1 KI mice. (**A**) Representative images of NeuN-positive neurons in 12-week and 30-week WT and SCA1 KI motor cortex, with insets of layers 2/3 and 5 (Scale bar top panel 200 μm, middle and bottom panel insets 50 μm), quantified in (**B**–**D**). (**B**), Quantification of cortical thickness in SCA1 KI and WT littermate controls at 12 and 30 weeks. (**C**,**D**), Quantification of NeuN-positive neuron number per mm^2^ (**C**) and nuclei area (**D**) in SCA1 KI and WT littermate controls at 12 and 30 weeks in cortical layers 2/3 and 5 (n = 3 animal per genotype, with 3 images per animal). (**E**), Representative images of Iba1-positive microglia in 12-week and 30-week WT and SCA1 KI motor cortex (scale bar top panel 200 μm, bottom panel inset 100 μm), quantified in (**F**–**I**). (**F**,**G**), Quantification of Iba1-positive microglia intensity (**F**) and number per mm^2^ (**G**) (*p*-value = 0.1339) in 12-week SCA1 KI cortex, compared to WT controls. (**H**,**I**), Quantification of Iba1-positive microglia intensity (**H**) and number per mm^2^ (**I**) in 30-week SCA1 KI cortex, compared to WT controls (n = 3 animal per genotype, with 3 images per animal). (**J**), Western blot images of Iba1 and Gfap protein expression in SCA1 KI and WT cortex at 12 weeks, with Vinculin (Vinc) as a loading control. (**K**,**L**), Quantification of Iba1 (**K**) and Gfap (**L**) protein expression levels from (**J**), normalized to Vinculin and WT expression levels, (n = 3 animal per genotype). (* *p* < 0.05, **** *p* < 0.0001, ns = not significant).

**Figure 2 cells-11-02632-f002:**
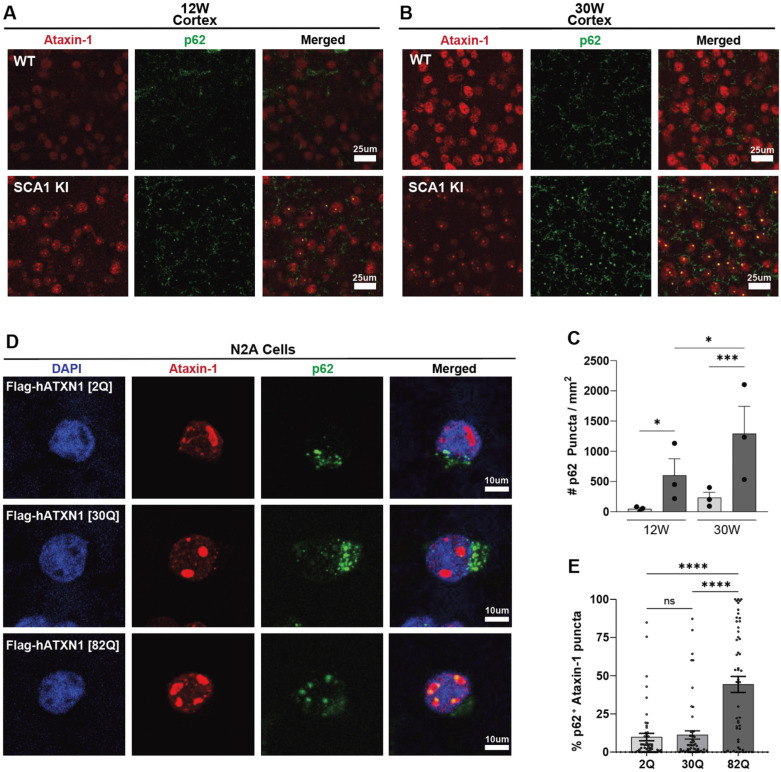
p62-positive aggregates are progressively upregulated and colocalize with ataxin-1 nuclear inclusions in the SCA1 KI mouse cortex. (**A**,**B**), Representative images of SCA1 KI and WT control motor cortex at 12-weeks (**A**) and 30-weeks (**B**), stained with ataxin-1 [11NQ] antibody (red) to label ataxin-1 nuclear inclusions and with p62 (green), scale bar 25 μm, quantified in (**C**). (**C**), Quantification of p62 puncta per mm^2^ in the WT and SCA1 KI mouse motor cortex and 12- and 30-weeks (n = 3 animal per genotype, with 3 images per animal). (**D**), Representative images of N2A cells transfected with human Flag-hATXN1 [2Q] (top row), Flag-hATXN1 [30Q] (middle row), and Flag-hATXN1 [82Q] (bottom row), stained with DAPI (blue), ataxin-1 [11NQ] (red), and p62 (green) (scale bar 10 μm), quantified in (**E**). (**E**), Quantification of p62-positive puncta colocalizing with ataxin-1 nuclear inclusions, plotted as % of cells with p62-positive ataxin-1 puncta, by polyQ length of *ATXN1* plasmid (n = 50 cells per condition). (* *p* < 0.05, *** *p* < 0.001, **** *p* < 0.0001, ns = not significant).

**Figure 3 cells-11-02632-f003:**
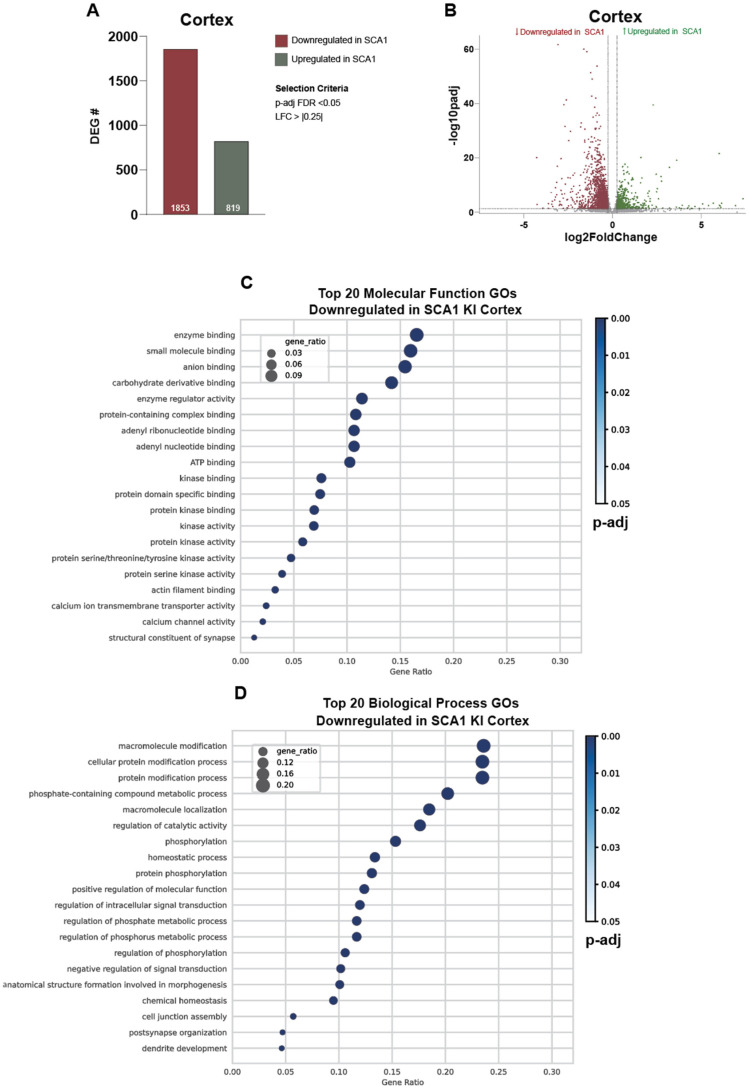
Transcriptomic alterations in the SCA1 KI mouse cortex at 12 weeks. (**A**), Number of up- and down-regulated DEGs in the SCA1 KI mouse cortex compared to WT littermate controls at 12 weeks. Genes with an FDR adjusted *p*-value < 0.05 and a log2fold change > |0.25| were considered significant. (**B**), Volcano plot of up- and down-regulated DEGs in the SCA1 KI mouse cortex, with significance cutoffs indicated. (**C**,**D**), Top 20 Molecular Function (**C**) and Biological Process (**D**) GO terms for down-regulated genes in the SCA1 KI mouse cortex. Top 20 GO terms are plotted by gene ratio, with circle color indicating adjusted *p*-value, and circle circumference indicating gene-ratio, or the ratio of DEGs identified within the listed GO term.

**Figure 4 cells-11-02632-f004:**
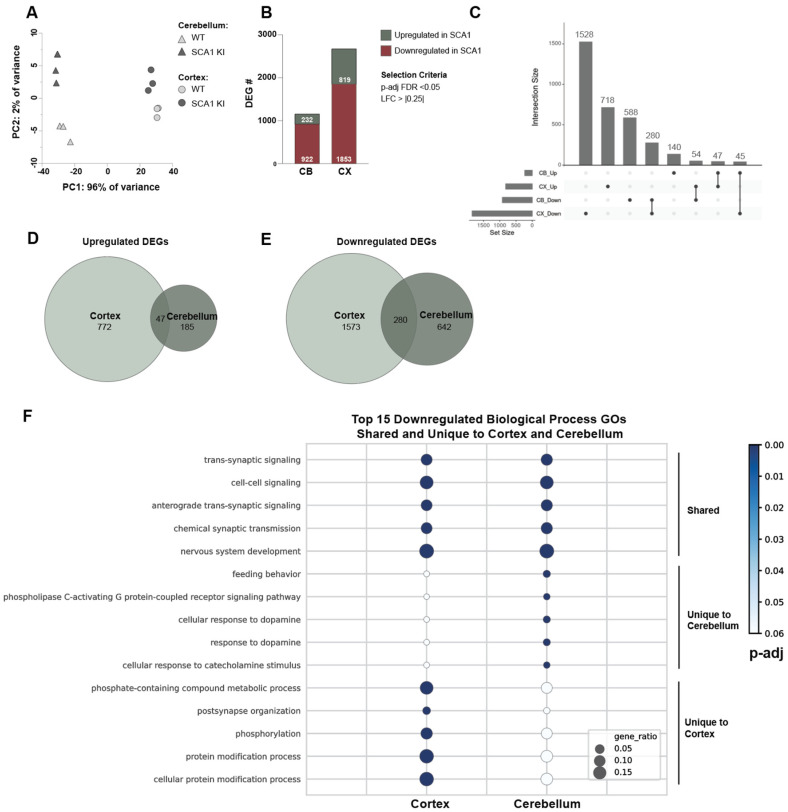
Regional comparison of shared and unique disease signatures between SCA1 KI mouse cerebellum and cortex at 12 weeks. (**A**), PCA plot of bulk RNA sequencing samples, showing samples cluster by region (shape) and genotype (color). (**B**), Number of DEGs that are up- and down-regulated in the SCA1 KI relative to WT for the cerebellum (CB) and cortex (CX). Genes with an FDR adjusted *p*-value < 0.05 and a log2foldchange > |0.25| were considered significant. (**C**), Upset plot showing number of up- and down-regulated DEGs shared across conditions. Set size indicates number of DEGs per condition, and intersection size indicates number of DEGs overlapping with indicated nodes. (**D**,**E**), Venn diagram indicating number of upregulated (**D**) and downregulated (**E**) DEGs that are unique to each region and shared across the cerebellum and cortex. (**F**), Top 15 Biological Process GO terms for downregulated DEGs in the SCA1 KI cerebellum and cortex that are shared across cerebellum and cortex (top), unique to the cerebellum (middle), and unique to the cortex (bottom). GO terms are plotted with circle color indicating FDR adjusted *p*-value, and circle size indicating gene ratio.

**Figure 5 cells-11-02632-f005:**
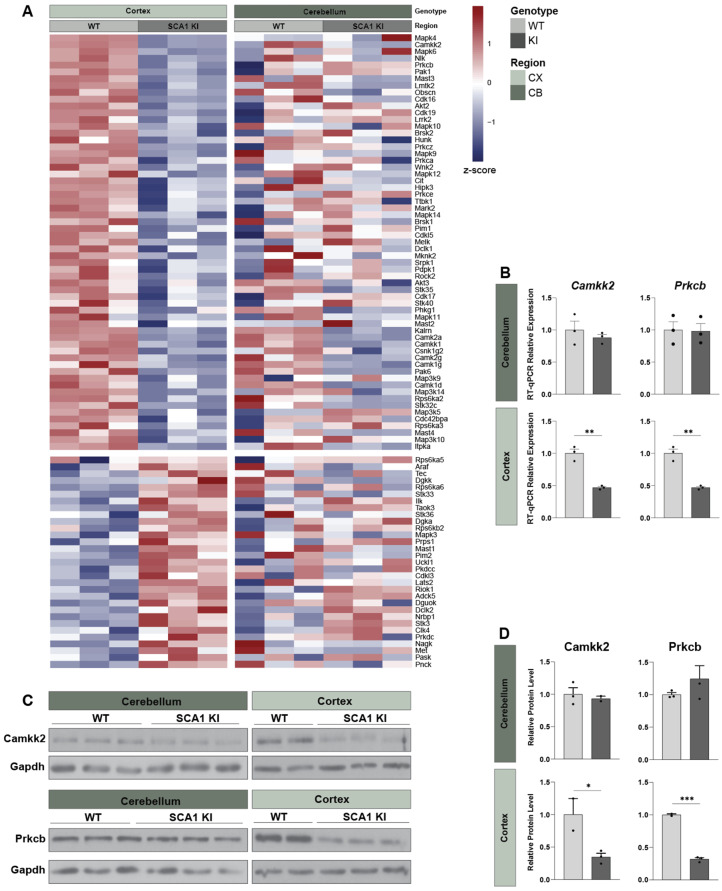
Cortex-specific downregulation of kinases in SCA1 KI mice. (**A**), Heatmap of downregulated (top panel) and upregulated (bottom panel) serine-threonine kinases (GO Term GO:0106310) in the SCA1 KI mouse cortex (left panel), relative to WT littermate controls, showing no consistent up- or down-regulation in the SCA1 KI mouse cerebellum (right panel). Cell color indicates Z-score scaled by row and region. (**B**), RT-qPCR validation of *Camkk2* and *Prkcb* mRNA expression levels in the 12-week SCA1 KI mouse cerebellum and cortex, normalized to housekeeping genes *Gapdh, Hprt,* and *ActB*, relative to WT littermate controls. (**C**), Western blot images of Camkk2 and Prkcb, with Gapdh as a loading control using the 12-week SCA1 KI mouse cerebellar or cortical protein extracts, quantified in (**D**). (**D**), Relative protein level of Camkk2 and Prkcb in the 12-week SCA1 cerebellum and cortex, normalized to Gapdh and relative to WT littermate controls (n = 3 per genotype). (* *p* < 0.05, ** *p* < 0.01, *** *p* < 0.001).

**Figure 6 cells-11-02632-f006:**
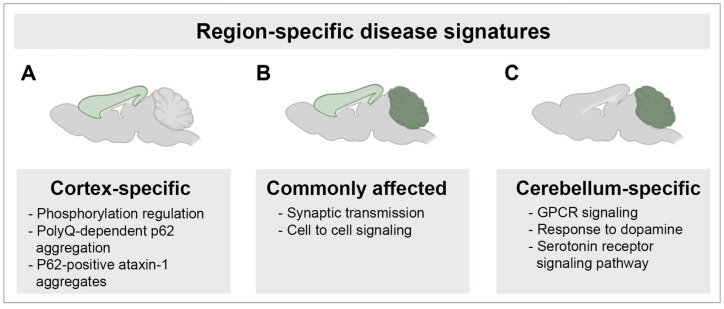
Schematic of shared and region-specific changes observed in the SCA1 KI mouse cortex and cerebellum during early stage of the disease at 12 weeks. (**A**), Cortex-specific changes observed in the SCA1 KI mice include region-specific differences in p62 aggregation colocalizing with ataxin-1 nuclear inclusions, and phosphorylation regulation. (**B**), Shared SCA1 disease signatures include synaptic transmission and cell to cell signaling. (**C**), Cerebellum-specific changes identified in the SCA1 KI mice include GPCR signaling, response to dopamine, and serotonin receptor signaling pathway.

## Data Availability

RNA sequencing data have been deposited in GEO under accession # (GSE211678).

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
