# Peer review of "Identifying Disease Signatures in the Spinocerebellar Ataxia Type 1 Mouse Cortex"

_cells, 2022, doi:10.3390/cells11172632_

Round 1
Reviewer 1 Report
The manuscript by Luttik et al aims to broaden the study of SCA1 disease in cortex. Previous work has looked at other brain regions, but this is the first investigation in cortex. The authors site as their rationale that other studies have demonstrated in humans with SCA1 there is degeneration and pathology in the frontal, temporal, and parietal lobes of the cortex on MRI and autopsy. Further in mouse models of SCA1 nuclear aggregates of polyglutamine ATXN1 are seen in cortical neurons. They highlight that the functional impacts of mutant atxn1 are not fully understood in the extra cerebellar brain regions, including the cortex. They cite that previous studies have shown region specific transcriptomic responses to the mutant Atxn1, specifically between cerebellum and inferior olive. The manuscript focuses on looking for the functional impact of mutant Atxn1 expression in cortex, studying the pathological process in this region and identifying it's unique transcriptomic response to the mutant Atxn1.
Summary of figures-
Figure 1: motor cortex is thin and more dense but total number of neurons doesn’t go down. No changes in astrocytes or microglia in this area at either time point.
Figure 2: atxn1 nuclear aggregates are seen in the motor cortex neurons and they co localize with p62. Supplemental figure 1 shows no colocalization of these two proteins in the cerebellum or brainstem. N2A cell culture experiments show that colocalization of p62 to atxn1 aggregates is CAG repeat length dependent.
Figure 3: RNAseq of cortex tissue, many DEGs observed.
Figure 4: Compare newly identified cortex DEG to previously identified cerebellar DEGs both at 12 weeks of age. Overall there were more DEGs in the cortex than cerebellum, some DEGs overlapped and a small subset went in the opposite direction between the 2 tissues. GO analysis suggest that synaptic dysfunction and neuron differentiation are shared between the 2 brain regions. In contrast kinase regulation was suggested to be uniquely downregulated and transcription factors upregulated in the cortex.
Figure 5: Validated by qPCR and WB that Camkk2 and Prkcb are specifically downregulated in SCA1 cortex and not cerebellum, highlighting prior work that different brain region specific kinases are altered in SCA1 pathogenesis.
Overall this is well written and thoughtful investigation of cortex pathophysiology in SCA1. It adds to the consensus in the field that brain region specific and/or cell type specific responses to the toxic polyglutamine ATXN1 may be unique and are not universal across the entire brain. This work adds important data to the field, that others can mine and compare to other regions of interest. There are some minor clarifications suggested below.
Minor points to consider addressing:
1) Please explain how the motor strip of the cortex was identified for the experiments, was it based on anatomical location on gross assessment of the brain or molecular markers on staining?
2) Along those same lines please be more precise in what is meant by “motor cortex”, I assume it's the motor strip in the frontal cortex where the upper motor neurons reside but there are other “motor” areas in the cortex, such as the prefrontal cortex.
3) N2A experiment, the 2Q and 30Q constructs don’t form atxn1 aggregates, the staining appears nucleolar and not bona fide aggregates, therefore the conclusion that the p62 aggregation is length dependent seems unfounded, instead I would conclude that p62 colocalizes to atxn1 when it is compartmentalized to the nuclear aggregates, and this is probably independent of how many repeats there are so long as there are enough repeats to trigger Atxn1 aggregation. Overall this experiment adds little to the paper and I would suggest removing it or moving it to supplemental data.
4) Is the bulk RNA seq of the whole cortex or motor cortex, please clarify in the text and if whole cortex provide rationale why this was used instead of just motor cortex where all the other studies are centered. It is confusing to me why the motor cortex is studied in figures 1-2 but then the rest of the paper focuses on the transcriptomic response in the entire cortex. I would suggest if this is the case that then a few other areas of the cortex are examined besides the motor cortex for figure 1 and 2, like maybe a slice through the temporal, parietal and occipital cortex so in the end you will have a representation of all four lobes of the brain cortex.
5) Validation of Camkk2 and Prkcb by qPCR and WB is robust but there is no mention of a rationale why these 2 kinases were chosen to validate, were others tried, were these the ones with the biggest fold change, ect….?
Reviewer 2 Report
In this report entitled “Identifying disease signatures in the spinocerebellar ataxia type 1 mouse cortex”, the authors specifically focus on the cortical signatures of disease both transcriptionally and biochemically. This work has led to novel signatures that are specific to the cortex as well as found conserved signatures that are consistently found in other vulnerable brain regions, like the cerebellum. Interesting among these novel signatures is colocalization of the p62 UPS/autophagy protein with ATXN1 in the cortex but not in the cerebellum or brainstem.
This work provides new opportunities for exploration of SCA1 disease pathogenesis in other vulnerable or possibly protective brain regions.
Overall, this is a comprehensive manuscript in which I agree with most of the conclusions and the significance of this work, although some minor clarifications and suggested data inclusions are required:
1. In the introduction, cognitive impairments reported in SCA1 patients is limited in description and references.
2. The rationale for the 12 and 30-week timepoints in not well defined. Are these presymptomatic and symptomatic timepoints as these relate to cognitive impairments in these mice?
3. In results section 3.1 paragraph three, second sentence implied GFAP cell counts were completed, yet Fig 1E-L does not show any supporting data.
4. In results section 3.2, P62 is referred to as marker of autophagy, however P62 also places an integral role in the ubiquitin proteosome system (UPS). Additionally, the interesting regional data of this protein is not well discussed within the context of both autophagy contributions and/or the UPS. Was there any supporting transcriptional data from the cortex that highlights one or both of these pathways in the cortex?
5. What is the rationale for the bulk RNAsequencing data completion only at 12 weeks of age in the cortex?
